# Compassionate Use Program of Ipilimumab and Nivolumab in Intermediate or Poor Risk Metastatic Renal Cell Carcinoma: A Large Multicenter Italian Study

**DOI:** 10.3390/cancers14092293

**Published:** 2022-05-04

**Authors:** Umberto Basso, Federico Paolieri, Mimma Rizzo, Ugo De Giorgi, Sergio Bracarda, Lorenzo Antonuzzo, Francesco Atzori, Giacomo Cartenì, Giuseppe Procopio, Lucia Fratino, Manolo D’Arcangelo, Giuseppe Fornarini, Paolo Zucali, Antonio Cusmai, Matteo Santoni, Stefania Pipitone, Claudia Carella, Stefano Panni, Filippo Maria Deppieri, Vittorina Zagonel, Giampaolo Tortora

**Affiliations:** 1Oncology 1 Unit, Department of Oncology, Istituto Oncologico Veneto IOV IRCCS, 35128 Padova, Italy; filippomaria.deppieri@aulss3.veneto.it (F.M.D.); vittorina.zagonel@iov.veneto.it (V.Z.); 2Medical Oncology Unit 2, Azienda Ospedaliera Universitaria Pisana, 43126 Pisa, Italy; federico.paolieri@gmail.com; 3Division of Translational Oncology, IRCCS Istituti Clinici Scientifici Maugeri, 27100 Pavia, Italy; rizzo.mimma@gmail.com; 4Department of Medical Oncology, IRCCS Istituto Romagnolo per lo Studio dei Tumori (IRST) “Dino Amadori”, 47014 Meldola, Italy; ugo.degiorgi@irst.emr.it; 5Medical and Translational Oncology Unit, Azienda Ospedaliera Santa Maria, 05100 Terni, Italy; s.bracarda@aospterni.it; 6Clinical Oncology Unit, Careggi University Hospital, 50135 Florence, Italy; lorenzo.antonuzzo@gmail.com; 7Department of Experimental and Clinical Medicine, University of Florence, 50121 Florence, Italy; 8Medical Oncology, Azienda Ospedaliero-Universitaria, 09124 Cagliari, Italy; francescoatzori74@yahoo.it; 9Oncology Unit, Azienda Ospedaliera di Rilievo Nazionale “Antonio Cardarelli”-AORN A. Cardarelli, 80131 Napoli, Italy; cartenigiacomo@gmail.com; 10Department of Medical Oncology, Fondazione IRCCS Istituto Nazionale dei Tumori, 20133 Milan, Italy; giuseppe.procopio@istitutotumori.mi.it; 11Medical Oncology Unit, Centro di Riferimento Oncologico CRO IRCCS, 33081 Aviano (PN), Italy; lfratino@cro.it; 12Department of Oncology and Hematology, Azienda Unità Sanitaria Locale della Romagna, 48121 Ravenna, Italy; mda20681@gmail.com; 13Medical Oncology Unit 1, Ospedale Policlinico San Martino IRCCS, 16132 Genova, Italy; giuseppe.fornarini@hsanmartino.it; 14Department of Biomedical Sciences, Humanitas University, 20090 Milan, Italy; paolo.zucali@hunimed.eu; 15Department of Oncology, IRCCS Humanitas Research Hospital, 20089 Milan, Italy; 16Don Tonino Bello Oncology, Istituto Tumori IRCCS Giovanni Paolo II, 70124 Bari, Italy; antoniocusmai@hotmail.com; 17Oncology Unit, Macerata Hospital, 62100 Macerata, Italy; mattymo@alice.it; 18Department of Oncology and Hematology, University of Modena and Reggio Emilia, Azienda Ospedaliero Universitaria of Modena, 41125 Modena, Italy; stefania.pipitone88@gmail.com; 19Medical Oncology Unit, Istituto Tumori IRCCS Giovanni Paolo II, 70124 Bari, Italy; claudia.carella@yahoo.it; 20Medical Oncology Unit, ASST di Cremona, 26100 Cremona, Italy; s.panni@asst-cremona.it; 21Università Cattolica del Sacro Cuore, 00168 Rome, Italy; giampaolo.tortora@policlinicogemelli.it; 22Fondazione Policlinico Universitario Agostino Gemelli—IRCCS, 00168 Rome, Italy

**Keywords:** ipilimumab, nivolumab, metastatic Renal Cell Carcinoma, toxicity, progression, survival, retrospective, immune-related adverse events

## Abstract

**Simple Summary:**

Ipilimumab and Nivolumab (IPI-NIVO) improved survival in a pivotal phase III trial conducted in patients with metastatic Renal Cell Carcinoma (mRCC) with intermediate or poor International Metastatic RCC Database Consortium (IMDC) score when compared to sunitinib. A compassionate use program of IPI-NIVO was launched in Italy from April through October 2019. The aim of our retrospective study was to assess the outcome of mRCC patients treated with IPI-NIVO within the Italian compassionate use program. Despite one third of patients not completing the induction phase of IPI-NIVO due to either toxicity or early progression, the 12-month survival rate of 66.8% confirms that IPI-NIVO combination is an effective first-line regimen for mRCC patients with IMDC intermediate-poor risk score.

**Abstract:**

This is a retrospective analysis on the safety and activity of compassionate Ipilimumab and Nivolumab (IPI-NIVO) administered to patients with metastatic Renal Cell Carcinoma (mRCC) with intermediate or poor International Metastatic RCC Database Consortium (IMDC) score as a first-line regimen. IPI was infused at 1 mg/kg in combination with Nivolumab 3 mg/kg every three weeks for four doses, followed by maintenance Nivolumab (240 or 480 mg flat dose every two or four weeks, respectively) until disease progression or unacceptable toxicity. A total of 324 patients started IPI-NIVO at 86 Italian centers. Median age was 62 years, 68.2% IMDC intermediate risk. Primary tumor had been removed in 65.1% of patients. Two hundred and twenty patients (67.9%) completed the four IPI-NIVO doses. Investigator-assessed overall response rate was 37.6% (2.8% complete). Twelve-month survival rate was 66.8%, median progression-free survival was 8.3 months. Grade 3 or 4 treatment-related adverse events occurred in 67 patients (26.9%). IMDC intermediate risk, nephrectomy, BMI ≥ 25 kg/m^2^, and steroid use for toxicities correlated with improved survival, while age < 70 years did not. IPI-NIVO combination is a feasible and effective regimen for the first-line treatment of intermediate-poor IMDC risk mRCC patients in routine clinical practice.

## 1. Introduction

Renal Cell Carcinoma (RCC) represents the 6th most frequently diagnosed cancer in men and the 10th in women, with an age-adjusted incidence ranging from 16.7 cases per 100,000 in the Czech Republic, to 12 in the United States, 8.7 in Italy, and less than 5 in South-Eastern Asia [1]. Clear cell histology is predominant, while papillary, cromophobe, or undifferentiated subtypes account for less than 20% of cases.

Immunotherapy has revolutionized the modern treatment of metastatic Renal Cell Carcinoma (mRCC) after the anti-Programmed Cell Death-1 monoclonal antibody Nivolumab (NIVO) was shown to improve overall survival (OS) and quality of life compared to everolimus in patients progressing after antiangiogenic regimens [2,3]. The association of NIVO with Ipilimumab (IPI), a monoclonal antibody targeting the cytotoxic T-Lymphocyte antigen-4 (CTLA-4), obtained durable responses in mRCC [4]. The pivotal CheckMate 214 phase III trial compared IPI-NIVO combination with sunitinib as a first-line regimen for mRCC patients [5]. Treatment consisted of 1 mg/kg of IPI combined with 3 mg/kg of NIVO, infused every three weeks for four cycles (induction phase), followed by NIVO 3 mg/kg every two weeks (maintenance phase), until disease progression or intolerable toxicity.

This study demonstrated improved OS and overall response rates (ORR) among intermediate and high International Metastatic RCC Database Consortium (IMDC) risk subgroups [6], with fewer symptoms and better health-related quality of life [7]. 

Immune-related adverse events (irAEs) of IPI-NIVO combination were more common than NIVO alone, with 46% of patients experiencing grade 3–4 toxicities [5] vs. 19%, respectively [2]. Twenty-one percent of patients did not complete the four IPI-NIVO cycles, and 35% of patients required high-dose glucocorticoids (≥40 mg of prednisone per day or equivalent) to treat adverse events. 

Since outcomes in terms of activity and—especially—toxicity of newer regimens applied to patients in real-world practice may differ from those reported in controlled clinical trials, it is of pivotal importance to collect safety and efficacy data as soon as these new regimens enter everyday clinical practice. This is necessary to demonstrate the generalizability of results issued by registration trials. 

This is of particular relevance for the IPI-NIVO regimen, which can achieve profound and long-lasting responses but can also cause severe and potentially life-threatening complications that are unpredictable and require prompt management with steroids and supportive care. General guidelines for the early recognition and treatment of irAEs have been issued and updated [8,9]. However, oncologists may need time to gain confidence in managing such toxicities during their initial experience with an IPI-NIVO regimen.

Following the approval of an IPI-NIVO combination for the first-line treatment of mRCC patients with intermediate-poor IMDC risk score by the Food and Drug Administration (FDA) and the European Medicines Agency (EMA), a compassionate use program of IPI-NIVO was launched in Italy from 1 April to 31 October 2019, allowing 86 oncology units to have early access to the IPI-NIVO combination for their patients with mRCC.

The aim of this study is to assess the tolerability and activity of the IPI-NIVO combination administered as a first-line regimen to patients with mRCC with an intermediate or poor IMDC risk score enrolled in the Italian compassionate use program.

## 2. Materials and Methods

This is a retrospective, non-interventional study of the clinical outcome of IPI-NIVO treatment in patients with mRCC. IPI and NIVO were provided by Bristol-Myers Squibb (BMS) upon physician request and after the approval by local Ethics Committees, in accordance with the Italian compassionate use regulation (Ministry of Health Decree, 7 October 2017). All patients signed an informed consent and privacy management form prior to commencing treatment. Eighty-six oncology units enrolled at least one patient each. This retrospective multicenter analysis was approved by the Ethics Committee of the Istituto Oncologico Veneto IOV IRCCS on 15 January 2021 (N 2022/139).

### 2.1. Inclusion Criteria

Inclusion criteria were the following: -Histological confirmation of RCC;-Eastern Cooperative Oncology Group (ECOG) performance status 0 to 2;-Advanced (not amenable to curative surgery or radiation therapy)/metastatic RCC Intermediate or poor IMDC score;-No prior treatment for metastatic disease;

Exclusion criteria were: -Prior treatment with checkpoint inhibitors as adjuvant therapy;-active or suspected autoimmune disease requiring the use of systemic immunosuppressive agents or steroids (>10 mg of prednisone or equivalents);-active (untreated) brain metastases requiring steroid therapy (>10 mg/day prednisone equivalents);-any other conditions requiring systemic treatment with corticosteroids (>10 mg daily of prednisone or equivalents) or other immunosuppressive medications within 14 days prior to the first dose of IPI-NIVO-any positive test result for hepatitis B or hepatitis C virus indicating presence of virus, e.g., Hepatitis B surface antigen (HBsAg) positive, or Hepatitis C antibody (anti-HCV) positive (except if HCV-RNA negative);-a known history of testing positive for human immunodeficiency virus (HIV) or known acquired immunodeficiency syndrome (AIDS);-major surgery (e.g., nephrectomy) less than 28 days prior to the first dose.

Inclusion and exclusion criteria were centrally reviewed by the BMS team on a dedicated website, where all patients were registered before authorizing the first supply of IPI-NIVO to the centers. The same platform was then used to manage the re-supply of IPI-NIVO and to notify treatment interruption. 

### 2.2. Treatment Regimen and Evaluation of Response

IPI was infused at a dose of 1 mg/kg plus NIVO 3 mg/kg, every three weeks for four doses, followed by maintenance NIVO (240 or 480 mg flat dose every 2 or 4 weeks, respectively) until progression, unacceptable toxicity, or withdrawal of consent.

Adverse events were monitored throughout the compassionate use program. These were graded using the National Cancer Institute Common Terminology Criteria for Adverse Events (CTCAE) system, version 5.0 [10]. 

Hematology, liver, and renal functions were checked before each drug administration. Radiological response was assessed by CT scan or MR scan. The first evaluation of response was planned after four cycles of IPI-NIVO and then per local clinical practice, approximately every 12 weeks. ORR was evaluated in accordance with Response Evaluation Criteria In Solid Tumors (RECIST 1.1) guidelines [11]. 

### 2.3. Statistical Analysis

Descriptive statistics were used to summarize patients’ clinical characteristics, comprising age, gender, performance status, body mass index (BMI), surgery on primary tumor, tumor pathology, sites of metastases, and IMDC score. 

OS was defined as the time from start of combination therapy to death from any cause. Whenever possible, survival of patients lost to follow-up was verified through municipal registries, otherwise they were censored at the last date known to be alive.

Progression-free survival (PFS) was calculated from the initiation of IPI-NIVO treatment until disease progression or death, whichever occurred first. 

OS and PFS were estimated according to the Kaplan–Meier method starting from the first day of therapy to the event. Survival outcomes were then evaluated in different prognostic subgroups according to IMDC risk score (intermediate vs. poor), number of IMDC factors in the Intermediate risk group (1 vs. 2), age (<70 vs. ≥70 years), cytoreductive nephrectomy (yes vs. no), BMI (<25 kg/m^2^ vs. ≥25 kg/m^2^) and use of steroids (yes vs. no). Groups were compared using the log-rank test. The association between baseline patient characteristics and efficacy outcomes was explored using Cox regression models. Risks were expressed as hazard ratios (HRs) with 95% confidence intervals (CIs). Statistical significance was considered if *p* < 0.05. All analyses were carried out using the Statistical Package for Social Science (SPSS) software.

## 3. Results

A total of 324 mRCC patients received at least one dose of IPI-NIVO within the Italian compassionate use program.

### 3.1. Patients’ Charactheristics

The patients’ characteristics are outlined in Table 1. Median age was 62 years, with 72 patients aged between 70 and 79 years and 14 ≥80 years. Most patients were males (74.1%) and asymptomatic before starting therapy (60.5%), but often had metastatic disease at diagnosis (61.7%). Ten patients presented with inactive autoimmune disease at the start of treatment (3%): thyroiditis, type 1 diabetes mellitus, vitiligo, and rheumatic polymyalgia. 

There was a prevalence of IMDC intermediate patients (68.2%) compared to poor risk patients (31.8%). Primary tumor had been removed in 65.1% of patients before treatment, while 8 additional patients underwent surgery after starting treatment. Clear cell histology was predominant (83.0%), followed by papillary (6.5%), chromophobe (2.5%), or unclassified (8.0%). Sarcomatoid features were reported in the pathology reports of 56 patients (17.3%). Eight percent of patients had brain metastases.

Some patients (16%) were receiving low doses of steroids (<10 mg of prednisone/prednisolone daily or equivalent) before starting therapy.

### 3.2. Early Interruption of Treatment

A total of 220 patients (67%) completed the four doses of IPI-NIVO, while 104 (32.1%) did not. Thirty-seven patients received only 1 cycle (11.4%). We compared the rate of early interruption of IPI-NIVO combination among the 13 centers that enrolled 7 or more patients compared to those that treated less than 7 patients, and found no difference (37.5% vs. 31.7%, *p* = 0.4). Causes of premature interruption of the combination therapy were disease progression in 49 patients (15.1%), toxicity in 55 patients (17.0%) (Table 2).

### 3.3. Overall Survival

After a median follow-up of 12 months (range 1–22 months), 117 patients have died (207 censored, 63.9%). Median OS was not reached (Figure 1), 12- and 18 month-survival rates are 66.8% and 57.3%, respectively.

OS of poor risk patients was significantly lower than the intermediate risk category, with 12-month OS at 43.2% vs. 77.2%, respectively (*p* < 0.0001, Figure 2).

Twelve-month OS rate was significantly different in intermediate risk patients with 1 IMDC risk factor (84.4%) compared to those with 2 IMDC risk factors (68.5%, *p* = 0.001). Patients who were older than 70 years appeared to have comparable OS to younger patients (12-month OS of 66.4% vs. 66.9%, *p* = 0.78) (Figure 3). 

Patients undergoing cytoreductive nephrectomy had superior OS compared to those who did not undergo surgery on their primary tumor (12-month OS of 74.2% vs. 51.2%, *p* < 0.0001) (Figure 4). 

Patients with a BMI ≥ 25 kg/m^2^ had better OS than those with a lower BMI (12-month OS of 72.3% vs. 62.3%, *p* = 0.03) (Figure 5). 

Moreover, 112 patients receiving steroids for the treatment of adverse events had a 12-month OS of 74.6% compared to 62.5% in untreated patients (*p* = 0.017) (Figure 6).

Patients who received less than four cycles of IPI-NIVO combination experienced shorter OS compared to those that completed the induction phase (12- and 18-month OS of 32.9 and 24.4% vs. 79.9 and 71.1%, respectively, *p* < 0.0001). Patients with non-clear cell histology experienced shorter OS compared to those with clear cell histology (12- and 18-month OS of 45.3 and 39.2% vs. 69.7 and 60.5%, respectively, *p* = 0.002). The presence of sarcomatoid differentiation showed a trend for reduced OS (12- and 18-month OS of 52.3 and 46.4% vs. 69.0 and 59.8%, respectively, *p* = 0.058)

### 3.4. Progression-Free Survival

As outlined in Figure 7, median PFS was 8.3 months (95% CI: 6.5–10.1 months, 122 censored, 37.7%).

Patients in the intermediate IMDC score group had a longer PFS compared to those in the poor risk category (10.4 vs. 2.9 months, *p* < 0.0001, Figure 8). 

Furthermore, PFS was significantly longer in intermediate risk patients with 1 IMDC risk factor (12.4 months) compared to those with 2 IMDC risk factors (7.9 months, *p* = 0.018).

Patients who interrupted early IPI-NIVO combination experienced shorter median PFS compared to the other patients (1.8 vs. 12.5 months, respectively, *p* < 0.0001). Patients with non-clear cell histology had shorter PFS compared to the other patients (3.4 vs. 10.2 months, respectively, *p* = 0.001). Patients with sarcomatoid differentiation had a median PFS of 8.2 months compared to 9.5 months in other patients (*p* = 0.73).

### 3.5. Response Rate

According to the investigators, 270 patients were able to be evaluated for radiological response after the IPI-NIVO induction phase. Five patients achieved a complete response (CR, 1.9%) and 76 a partial response (PR, 28.1%) for an ORR of 30.0%; 96 patients had stable disease (SD, 35.6%) and 93 had progression (PD, 34.4%) as their best response to the IPI-NIVO combination (Table 3). Patients in the intermediate IMDC risk category had an ORR of 28.1% compared to 18.4% in the poor risk group, although the difference was not statistically significant (*p* = 0.24).

A total of 213 patients started maintenance NIVO (65.7%), with median number of cycles being 10 (range 1–37). The number of patients evaluated for response rose to 282, and 25 additional responses (3 complete and 22 partial) were reported by the investigators. Thus, the overall ORR to treatment was 37.6% (34.8% partial and 2.8% complete) (Table 3).

Patients who received less than four cycles of IPI-NIVO combination achieved a lower ORR compared to those who completed the four cycles (12.5% vs. 41.8%, respectively, *p* < 0.0001). Non-clear cell patients showed a reduced ORR (18.2%) compared to those with clear cell histology (35.3%, *p* = 0.013). Conversely, the presence of sarcomatoid differentiation did not appear to influence ORR (33.9% vs. 32.1% in non-sarcomatoid patients (*p* = 0.79).

### 3.6. Safety 

Treatment with IPI-NIVO was interrupted early in a total of 55 patients (17.0%) due to toxicity. Treatment-related adverse events (irAEs) causing premature interruption of IPI-NIVO are listed in Table 4. The most frequent were gastrointestinal events, followed by hepatic, pulmonary, pancreatic, or neurological toxicities. A total of 112 patients (34.6%) received steroids, 53 of whom (16.4%) received doses ≥40 mg/day of prednisone or prednisolone. Four patients (1.2%) required second-level immunosuppressive drugs (mycophenolate, infliximab, or other drugs), whereas four patients (1.2%) died from severe adverse events: two from gastro-intestinal toxicity, one from liver impairment and one from renal failure.

Out of 213 patients starting maintenance nivolumab, 93 (28.7% of the total population) were still on treatment at the cut-off date of 1 March 2021.

The overall toxicity of treatment is outlined in Table 5. The worst toxicities were grade 1–2 in 62.7% of patients and grade 3 or 4 in 26.9%.

### 3.7. Second-Line Therapy

A total of 108 patients started a second line therapy. Cabozantinib (53.7%) and sunitinib (39.8%) were the most frequent options. Fewer patients started pazopanib (2.8%) or other therapies. Out of 82 evaluated patients, 2 complete (2.4%) and 25 partial responses (30.5%) were reported, for an overall response rate of 32.9%. Thirty patients had stable disease (36.6%) and 25 progression (30.5%) as their best response to second-line regimens.

## 4. Discussion

The IPI-NIVO combination improved OS compared to either nivolumab or ipilimumab monotherapies in metastatic melanoma [12]. At the same time, IPI-NIVO in association with chemotherapy improved survival in patients with lung cancer [13]. Further studies in other tumor types are ongoing.

The pivotal phase III CheckMate 214 trial demonstrated the superiority of the IPI-NIVO combination over sunitinib in first-line treatment of mRCC patients with intermediate and poor IMDC risk scores [5]. Updated results at 5-year follow-up were recently reported at the 2021 European Society of Medical Oncology Congress [14]. When considering only the intermediate and poor IMDC risk cohort, median OS with IPI-NIVO was significantly longer compared to sunitinib (47.0 months vs. 26.6 months, respectively) with an HR of 0.68, *p* < 0.0001. Significant benefits in terms of PFS (median 11.2 vs. 8.3 months, respectively, HR 0.73, *p* = 0.0004) and ORR (42% vs. 27%) were also reported, with an overall complete response rate of 11% with IPI-NIVO compared to 2% with sunitinib. A safety analysis confirmed that fewer grade 3–4 treatment-related AEs were experienced by patients in the NIVO + IPI arm (48%) compared to sunitinib (64%), although treatment-related adverse events leading to discontinuation of therapy occurred more often with IPI-NIVO (23% vs. 13%).

A detailed analysis of patient-reported outcomes (PROs) was conducted during the study, demonstrating that treatment with IPI-NIVO preserves a better health-related quality of life compared to sunitinib, with significant differences found for four of five Functional Assessment of Cancer Therapy Kidney Symptom Index-19 (FKSI-19) domains (disease-related symptoms, physical disease-related symptoms, treatment side-effects, and functional wellbeing) and Functional Assessment of Cancer Therapy-General (FACT-G) physical and functional wellbeing domains [7]. 

Following FDA and EMA approval of the IPI-NIVO combination for the treatment of mRCC, a compassionate use program was launched in Italy while waiting for full regulatory approval. These early access programs are of paramount importance in order to test innovative treatment regimens in a wider population of real-world patients, who are notoriously less selected compared to fit patients enrolled in registration clinical trials. 

The present study is a retrospective analysis of safety and activity of IPI-NIVO combination in 324 mRCC patients enrolled within the Italian compassionate use program. Our study population had the same median age (62 years) as the CheckMate 214 study [5], but included a higher proportion of patients with a poor IMDC risk score (32% vs. 21%) and no previous nephrectomy (34.9% vs. 20%). As for the sites of metastases, the prevalence of lung (71.3% vs. 69%) and liver metastases (18.2% vs. 21%) was fairly comparable, but we found a higher proportion of bone metastases (30.9% vs. 22%), and we also enrolled 26 patients (8%) with brain metastases who were excluded from the registration trial. 

Only RCC with a clear-cell component was included in the randomized trial. On the contrary, this real-world study population comprised 17% of patients with non-clear cell histologies. When analyzed separately, these non-clear cell patients experienced a shorter PFS (3.4 months), OS (45% at 12 months), and lower response rates compared to those with clear cell histology, in line with the results recently reported with IPI-NIVO combination in the phase IIIb/IV CheckMate 920 trial (median PFS and OS of 3.7 and 21 months, respectively) [15]. 

Selection bias due to adverse prognostic factors may therefore explain the lower rates of 12- and 18-month OS reported in our study (66.8% and 57.3%) compared to the Checkmate 214 trial [5] (80 and 75%, respectively), as well as the shorter median PFS (8.3 vs. 11.6 months). Instead, our results are comparable to those registered in a real-world cohort of 191 Canadian mRCC patients treated with IPI and NIVO, with 12-month OS of 72.2% and median PFS of 7.4 months [16]. 

Patients who interrupted the IPI-NIVO induction phase early achieved a shorter PFS (1.8 months) and OS (32.9% at 12 months) compared to those who completed the four cycles. A very low 4-month OS probability (36%) was recently reported in a real-world cohort of intermediate or poor IMDC risk patients starting IPI-NIVO after hospitalization for mRCC-related symptoms, as a demonstration that patients with more advanced and symptomatic disease derive less benefit from the IPI-NIVO combination [17]. Therefore, a longer follow-up of our cohort will be of paramount importance to evaluate the rate and clinical characteristics of long-term responders—the group of patients deriving the greatest benefit from IPI-NIVO synergistic activation of the immune response. 

As expected, patients with an intermediate IMDC risk assessment showed higher OS and PFS compared to the poor risk category. Interestingly, the subgroups of intermediate patients who had only one adverse IMDC prognostic factor fared significantly better than those with two adverse factors, both in terms of OS (12-month OS 84.4% vs. 68.5%, *p* < 0.0001) and PFS (median 10.4 vs. 2.9 months, *p* < 0.001), as already reported in patients treated with anti-angiogenic drugs [18,19], allowing us to better refine our ability to discriminate prognosis of patients.

Older patients in our cohort achieved a comparable OS to younger patients, as reported in the subgroup analyses of CheckMate 214 [5], CheckMate 069 [12], and CheckMate 9LA [13]. This is a further confirmation that the efficacy of immunotherapy is not influenced by age, including genitourinary tumors [20]. Yet, it must be remembered that older people require more careful observation for the management of adverse events [21]. Prospective studies on the application of G8 and modified-G8 screening tools for multidimensional geriatric assessment before administration of immunotherapy to RCC patients are therefore warranted.

The randomized CARMENA trial failed to demonstrate a positive effect of cytoreductive nephrectomy in mRCC patients treated with sunitinib [22]. Conversely, patients receiving nephrectomy in our cohort had a significantly better survival than patients retaining their primary renal tumor [12-month OS of 74.2% vs. 51.2%, respectively *p* < 0.0001] as other investigators recently reported with the use of immunotherapy in mRCC [23]. Whether selection bias or relevant biological mechanisms may cause such difference has still to be determined: randomized controlled trials such as the PROBE trial (NCT04510597) are ongoing.

It has been reported that a higher BMI is correlated with improved survival after Nivolumab administration in mRCC patients [24]. Indeed, when compared to a lower BMI (12-month OS 72.3% vs. 62.3%, respectively *p* = 0.03), having a BMI ≥ 25 kg/m^2^ correlated with improved survival in our cohort also. Several authors reported that, although obesity may induce a state of chronic low-grade inflammation which may impair effector immune populations, it may also correlate with improved benefit on patient survival following anti-PD-1 or anti-CTLA-4 treatment [25]. These data suggest that nutritional counseling and the prevention of sarcopenia should be further explored in mRCC patients treated with immunotherapy. In our cohort, radiological response was evaluated by the investigators and was not reviewed centrally. 

In our study the ORR and PFS of patients with sarcomatoid differentiation appeared to be comparable to the non-sarcomatoid patients, in line with recent reports showing increased benefit from immunotherapy in these prognostically disadvantaged patients, whose outcome is particularly dismal when treated with tyrosine kinase inhibitors (TKIs) alone [26].

The overall response rate of 37.6% is slightly inferior compared to the centrally reviewed rate of 42.1% achieved by the IPI-NIVO combination in the Checkmate 214 trial [27], with fewer complete responses (2.8% vs. 10.1%) and a higher rate of progression as best response (34.0 vs. 19.8%). A PD-L1 positivity > 1% of cells correlated with increased response rates in the Checkmate 214 trial [5], but PD-L1 expression could not be tested in pathological samples from our population. 

A higher rate of our patients (32.1%) compared to the registration trial (21%) did not complete the four IPI-NIVO induction cycles because of adverse events or early progression. Apart from a negative selection bias, we may also hypothesize that the limited experience of smaller centers with the management of IPI-NIVO toxicities and evaluation of response to immunotherapy might have induced a premature withdrawal of treatment in some patients. Yet, when we compared the early interruption rate among centers treating more or less than seven patients, no differences were found. 

The most frequent adverse events leading to early interruption of IPI-NIVO were those typically associated with this regimen in mRCC and other tumors: gastrointestinal, hepatic, pulmonary, or pancreatic toxicities [5,12,13]. The overall rate of grade 3–4 irAEs with IPI-NIVO reported by the investigators in our cohort was 26.9%, which was inferior to that of the CheckMate 214 study (46%) [5] and of the CheckMate 067 in melanoma patients (59%) [12]. However, toxicity was assessed retrospectively and may therefore have been underreported. 

The use of high-dose glucocorticoids (≥40 mg/daily of prednisone/prednisolone) to treat irAEs was 16.4% vs. 29% in the Checkmate 214 trial [5]. Similar to what has already been reported in melanoma and other tumors [12,28], the use of steroids to treat irAEs from check-point inhibitors did not appear to worsen OS in our study. Instead, a survival advantage was registered for treated versus untreated patients (12-month OS rate of 74.6% vs. 62.5%, respectively; *p* = 0.017). Even in the era of cytokines, there had been reports of correlation between incidence of adverse events from interleukin-2 and interferon and improved treatment benefit in mRCC patients [29]. Several reports have since been published that demonstrate a correlation between the incidence of irAEs and improved benefits with immunotherapy with checkpoint inhibitors not only in melanoma patients [30] but also in those with mRCC treated with NIVO monotherapy [31] or IPI-NIVO combination [32]. As a whole, we registered a treatment-related death rate of 1.2% (4 patients), which is comparable to the rate reported in the IPI-NIVO arm of the CheckMate 214 [5] (8 patients, 1.5%) and the CheckMate 067 study (2 out of 314 patients) [12]. 

In more recent years, several studies have tested the association of anti PD-1 antibodies with TKIs, achieving impressive results in terms of increased response rates and survival benefits compared to sunitinib monotherapy and regardless of PDL-1 expression, leading to rapid approval by the FDA and EMA for all IMDC risk groups. Combined regimens of pembrolizumab plus axitinib [33], avelumab plus axitinib [34], nivolumab plus cabozantinib [35], and pembrolizumab plus lenvatinib [36] are currently alternative options to IPI-NIVO combination for the first-line treatment of mRCC patients with intermediate and poor IMDC risk score. Some authors have tried to categorize which clinical factors (performance status and symptoms, site and extension of metastases) or pathological aspects (histology, PD-L1 expression, molecular markers of angiogenesis, and immune evasion) might guide oncologists in the choice between an immune–immune combo over an immune–TKI regimen [37,38]. 

To date, IPI-NIVO treatment appears to be more suitable for older and obese patients as well as those with cardiac problems due to their increased risk of toxicities from TKIs; and in patients affected by RCC with sarcomatoid features and those with IMDC intermediate-risk and low disease burden (especially lymphnodes), in whom long-lasting complete responses can be achieved.

A randomized trial (COSMIC-313, NCT03937219) comparing the association of IPI-NIVO plus cabozantinib against an IPI-NIVO doublet has already completed accrual and will determine if triplets might become the best option in the future. 

Very recent data have shown that anti-Programmed Cell Death-1 therapy with pembrolizumab reduces disease relapses after surgery in patients with localized disease [39], thus opening the way for future studies of immunotherapy combinations also in the postoperative and neoadjuvant settings.

## 5. Conclusions

This compassionate use program enrolled a population of mRCC patients with an increased prevalence of adverse prognostic factors compared to registration in the Checkmate 214 trial, as well as patients with brain metastases who were excluded from the same study. With the limitations of selection bias and retrospective design, OS, PFS, and response rates appear lower compared to the registration trial, with a slightly higher number of patients who did not receive the four doses of the IPI-NIVO combination due to adverse events or early progression. However, the number of treatment-related deaths appears superimposable and very few patients needed a second-level immunosuppressive therapy. 

This compassionate use program confirms that IPI-NIVO is an effective combination for the treatment of metastatic RCC in routine clinical practice. 

In Italy, since January 2022, the IPI NIVO combination has been approved and reimbursed for first-line treatment of mRCC patients with an intermediate-poor IMDC risk score. Further studies will be needed to discern which patients should receive this association of anti PD-1/anti CTLA-4 combination or alternatively the novel combinations of TKIs plus anti-PD-1 antibodies.

## Figures and Tables

**Figure 1 cancers-14-02293-f001:**
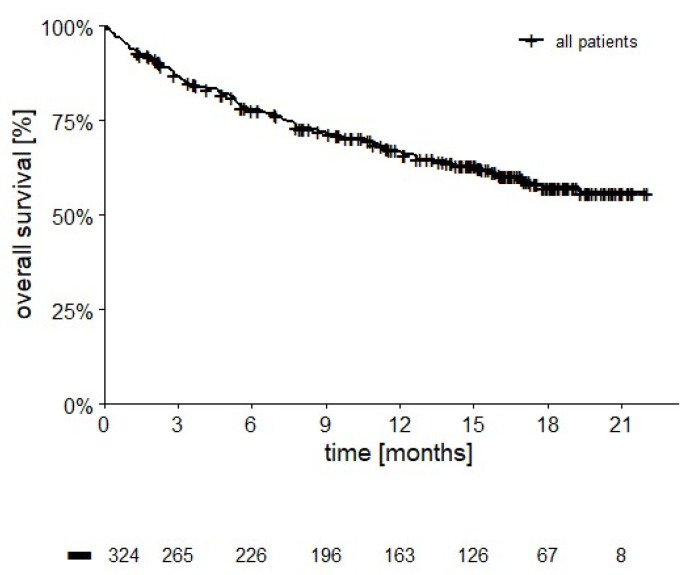
OS in 324 patients (median not reached, 12-month OS = 66.8%; 18-month OS = 57.3%, 207 censored, 63.9%).

**Figure 2 cancers-14-02293-f002:**
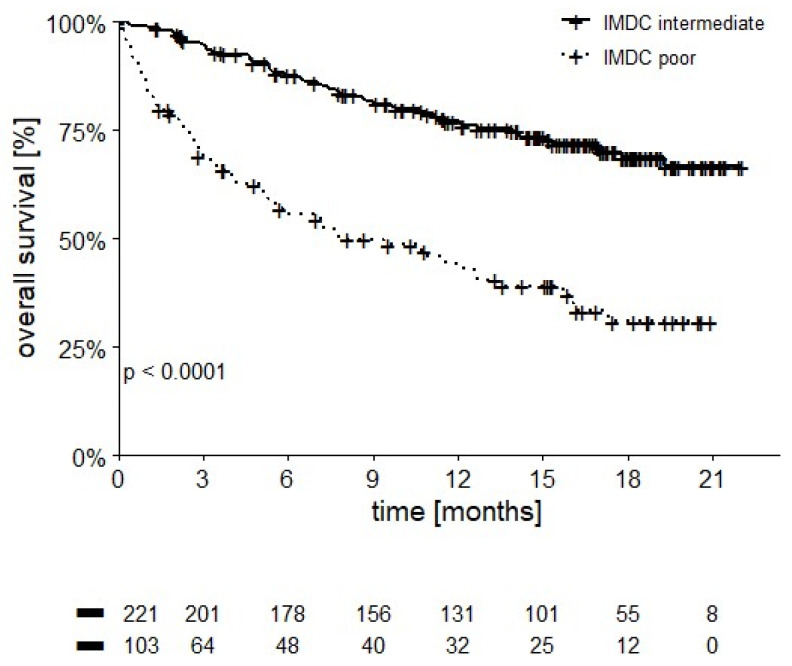
OS stratified by IMDC score (12-month OS 43.2% in poor vs. 77.2% in intermediate IMDC patients, *p* = 0.001).

**Figure 3 cancers-14-02293-f003:**
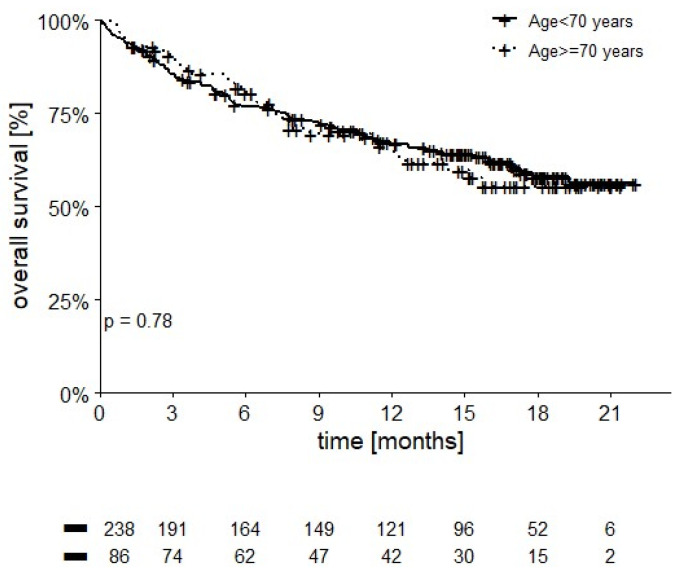
OS stratified by age (12-month OS 66.4% in patients ≥ 70 years vs. 66.9% in younger patients, *p* = 0.78).

**Figure 4 cancers-14-02293-f004:**
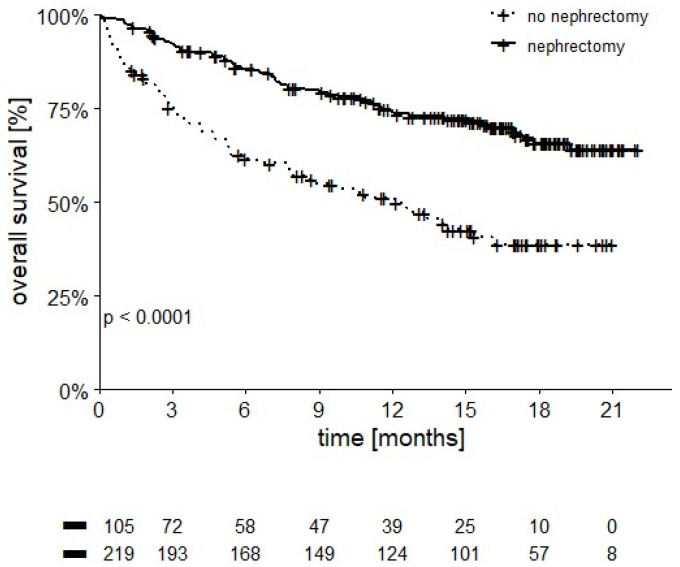
OS stratified by nephrectomy (12-month OS 74.2% in patients undergoing nephrectomy vs. 51.2% in patients who were not, *p* ≤ 0.0001).

**Figure 5 cancers-14-02293-f005:**
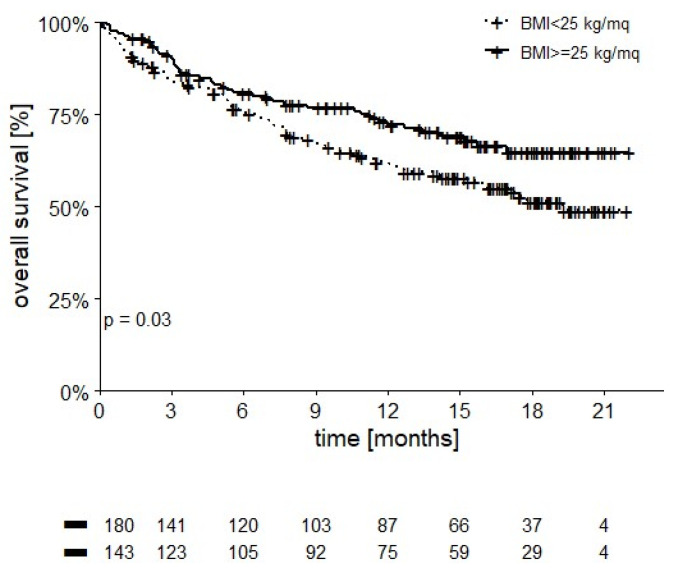
OS stratified by BMI (12-month OS 72.3% in patients with BMI ≥ 25 kg/m^2^ vs. 62.3% in patients with lower BMI, *p* = 0.03).

**Figure 6 cancers-14-02293-f006:**
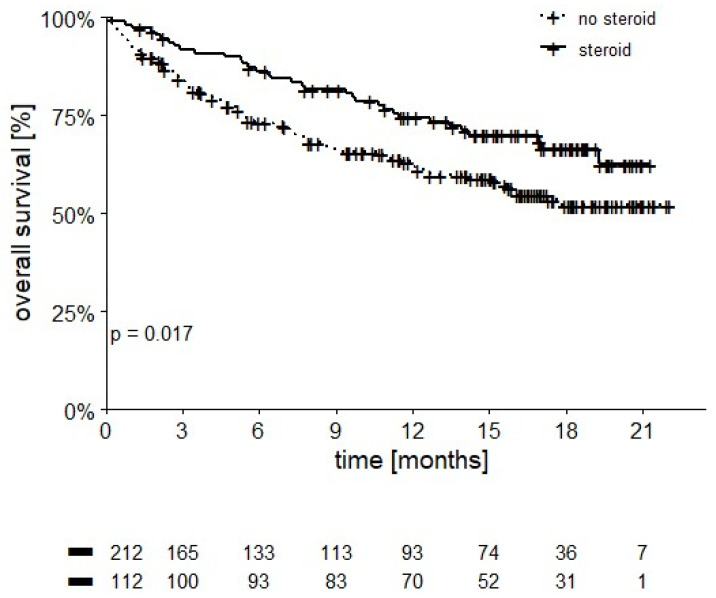
OS stratified by steroid use (12-month OS of 74.6% in patients receiving steroids vs. 62.5% in those who were not, *p* = 0.017).

**Figure 7 cancers-14-02293-f007:**
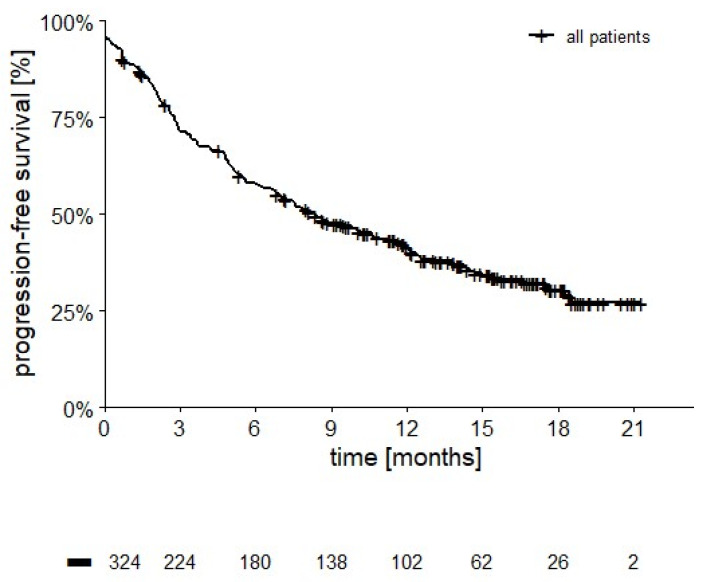
PFS in 324 patients (median PFS 8.3 months (95% CI: 6.5–10.1 months, 122 censored, 37.7%).

**Figure 8 cancers-14-02293-f008:**
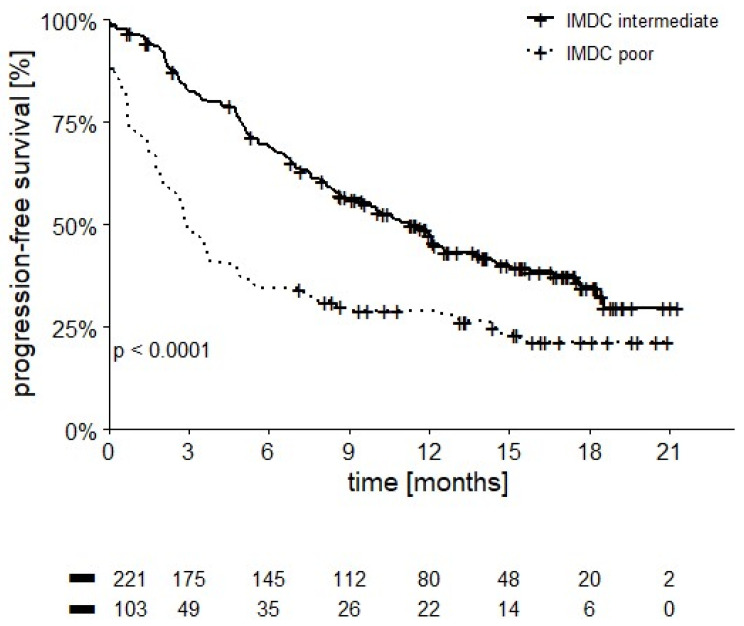
PFS in intermediate and poor IMDC patients (median PFS 10.4 vs. 2.9 months, *p* < 0.0001).

**Table 1 cancers-14-02293-t001:** Patients’ characteristics.

	*n* = 324		
**Age, median (range)** **70–79 years; ≥80 years**	62 years (24–87) 72 pts (22.5%)/14 pts (4.3%)		
**Gender**	240 Males (74.1%)/84 Females (25.9%)		
		n	*%*
**Symptoms**	yes	128	39.5
	no	196	60.5
**ECOG PS**	0	177	55
	1	115	36
	2	32	10
**BMI**	≥25 kg/m^2^	143	44.1
	<25 kg/m^2^	181	55.9
**Autoimmune disease**	yes	10	3.1
	no	314	96.9
**Nephrectomy**	before treatment	211	65.1
	during treatment	8	2.5
	None	105	32.4
**Histology**	clear cell	269	83.0
	papillary	21	6.5
	chromophobe	8	2.5
	unclassified	26	8.0
**WHO grade**	G1-2	48	14.8
	G3-4	130	40.1
	NA	146	45.1
**Metastases at diagnosis**	yes	200	61.7
	no	95	29.3
	undefined	29	9
**Sites of metastases**	lung	231	71.3
	bone	100	30.9
	liver	59	18.2
	brain	26	8.0
	pancreas	15	4.6
	other	80	24.7
**IMDC risk score**	Intermediate	221	68.2
	1 risk factor	119	36.7
	2 risk factors	102	31.5
	poor	103	31.8

Legend: ECOG PS: Easter Cooperative Oncology Group Performance Status, BMI: body mass index, WHO: World Health Organization; IMDC: International Metastatic Database Consortium.

**Table 2 cancers-14-02293-t002:** Administration of 4 cycles of IPI-NIVO.

	*n* = 324	
**Number of cycles of IPI-NIVO received**	*n*	%
1	37	11.4
2	33	10.2
3	34	10.5
4 (as planned)	220	67.9
**Reason for early discontinuation**	** *n* **	**%**
Progression	49	15.1
Adverse events	55	17.0

**Table 3 cancers-14-02293-t003:** Investigator-assessed response rate (270 patients evaluable after IPI-NIVO, and 282 evaluable after start of nivolumab maintenance therapy).

	First Evaluation after IPI-NIVO	Additional Responses during Nivolumab Monotherapy	Overall Response Rate
Response	Patients (%)	Patients (%)	Patients (%)
Complete response	5 (1.9)	3 (1.1)	8 (2.8)
Partial response	76 (28.1)	22 (7.8)	98 (34.8)
Stable	96 (35.6)	-	80 (28.4)
Progression	93 (34.4)	-	96 (34.0)
Total evaluable	270 (100)	-	282 (100)

**Table 4 cancers-14-02293-t004:** Immune-related Adverse Events (irAEs) causing early interruption of IPI-NIVO. *n* = 55, two deaths due to gastrointestinal toxicity *, one due to hepatic toxicity ^†^, and one due to renal toxicity ^#^.

AE	*n* = 55	%
Gastrointestinal	18 *	32.7
Hepatic	8 ^†^	14.5
Pulmonary	5	9.1
Pancreatic	5	9.1
Neurological	4	7.3
Asthenia	2	3.6
Cardiac	1	1.8
Endocrine	1	1.8
Allergic event	1	1.8
Skin toxicity	1	1.8
Renal	1 ^#^	1.8
Anemia	1	1.8
Others	12	21.8

**Table 5 cancers-14-02293-t005:** Cumulative incidence of worst immune-related adverse events (irAEs).

irAE	Grade 1–2 (%)	Grade 3–4 (%)	Grade 5 (%)
Gastrointestinal	39 (12.0)	21 (6.5)	2(0.6)
Hepatic	12 (3.7)	16 (4.9)	1 (0.3)
Pulmonary	4 (1.2)	8 (2.5)	0
Pancreatic	4 (1.2)	9 (2.8%)	0
Neurological	13 (4.0)	6 (1.9)	0
Asthenia	15 (4.6)	2 (0.6)	0
Cardiac	-	1 (0.3)	0
Thyroid	51 (15.7)	4 (1.2)	0
Hypophysitis	5 (1.5)	5 (1.5)	0
Skin	54 (16.7)	5 (1.5)	0
Anemia	3 (0.9)	2 (0.6)	0
Others	3 (0.9%)	8 (2.5%)	1 (0.3%)
Total	203 (62.7)	67 (26.9%)	4 (1.2%)

## Data Availability

The data are not publicly available due to privacy.

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
