# Peer review of "Compassionate Use Program of Ipilimumab and Nivolumab in Intermediate or Poor Risk Metastatic Renal Cell Carcinoma: A Large Multicenter Italian Study"

_cancers, 2022, doi:10.3390/cancers14092293_

Round 1

Reviewer 1 Report

The current study demonstrates the compassionate use of the IPI-NIVO combination to have a 66.8% survival rate over 12-months in a retrospective non-interventional study involving metastatic renal cell carcinoma (mRCC) patients with intermediate or poor International Metastatic RCC Database Consortium (IMDC) risk score.

The references are appropriate, with more than 90% of the cited references within the last five years. In addition, the references do not have any abnormal self-citations by the authors. The Year of publication for reference number 10 seems to be missing.

The article contributes knowledge to the current field, demonstrating that poor risk patients' OS over 12-month was 34% less along with decreased PFS compared to the intermediate-risk category (IRC). Similarly, IRC with 1 IMDC risk factor had 15.4% better OS and longer PFS than 2 IMDC risk factors. 12 and 18 months OS was either lower compared to the Checkmate 214 trial or comparable to the Canadian real-world cohort.

The tables and figures in the article provide valuable information and a comparison of the data.

A long-term follow-up and evaluation of the cohort will provide additional relevant information about the benefit of the IPI-NIVO combination.

The authors have mentioned that one limitation related to premature withdrawal of patients is the limited experience of smaller centers in managing IPI-NIVO toxicities and evaluating response to immunotherapy. The authors may as well provide comparative data on the patients that have prematurely withdrawn from all the centers to confirm this hypothesis.  

Author Response

Reviewer 1
(x) English language and style are fine/minor spell check required.

The revised version has been reviewed and checked for spelling mistakes and typing errors.

The current study demonstrates the compassionate use of the IPI-NIVO combination to have a 66.8% survival rate over 12-months in a retrospective non-interventional study involving metastatic renal cell carcinoma (mRCC) patients with intermediate or poor International Metastatic RCC Database Consortium (IMDC) risk score.

 The references are appropriate, with more than 90% of the cited references within the last five years. In addition, the references do not have any abnormal self-citations by the authors. The Year of publication for reference number 10 seems to be missing.

The year of reference number 10 is 2009, as now added to the text

The article contributes knowledge to the current field, demonstrating that poor risk patients' OS over 12-month was 34% less along with decreased PFS compared to the intermediate-risk category (IRC). Similarly, IRC with 1 IMDC risk factor had 15.4% better OS and longer PFS than 2 IMDC risk factors. 12 and 18 months OS was either lower compared to the Checkmate 214 trial or comparable to the Canadian real-world cohort.

The tables and figures in the article provide valuable information and a comparison of the data.

A long-term follow-up and evaluation of the cohort will provide additional relevant information about the benefit of the IPI-NIVO combination.

A collection of updated clinical data is planned for 2022 and a new article is scheduled for 2023.

The authors have mentioned that one limitation related to premature withdrawal of patients is the limited experience of smaller centers in managing IPI-NIVO toxicities and evaluating response to immunotherapy. The authors may as well provide comparative data on the patients that have prematurely withdrawn from all the centers to confirm this hypothesis.  

We compared the rates of early interruption of IPI-NIVO combination  among the 13 Centers that enrolled 7 or more patients compared to those  that treated 6 or less patients, and found  no difference (37.5 vs 31.7%, p=0.4). Patients who received less than four cycles of IPI-NIVO combination experienced shorter OS (12- and 18-months-OS of 32.9  and 24.4% vs 79.9 and 71.1%, respectively, p<0.0001), shorter PFS (median of 1.8 vs 12.5 months, p<0.001) and lower ORR (12.5 vs 41.8%, p<0.0001).

Reviewer 2 Report

This is a very well-written study describing the real-world usage of ipilimumab and nivolumab in metastatic RCC in Italy.

I have one important recommendation for this article: One major interest is the use of these ICIs in non-clear cell RCC. In this study, 17% of the patients are non-ccRCC (6.5% papillary, 2.5% chromophobe, 8.0% unclassified). If available, more data on these patients should be displayed in the results section including PFS and OS, comparing them to ccRCC. I believe this would strengthen the manuscript significantly. In the discussion, I would also recommend a paragraph or two about the non-ccRCC cohort.

Some English editing is required. For example, Authors and Investigators do not need to be capitalized. Kg in line 76 does not need to be capitalized. There are also several awkward double spaces between two words throughout the text.

Line 45 should say "Despite one third of patients not completing..."

Make sure to define TKI before using it (line 481).

Author Response

This is a very well-written study describing the real-world usage of ipilimumab and nivolumab in metastatic RCC in Italy.

I have one important recommendation for this article: One major interest is the use of these ICIs in non-clear cell RCC. In this study, 17% of the patients are non-ccRCC (6.5% papillary, 2.5% chromophobe, 8.0% unclassified). If available, more data on these patients should be displayed in the results section including PFS and OS, comparing them to ccRCC. I believe this would strengthen the manuscript significantly.

Patients with non-clear histology achieved lower OS, PFS and ORR compared to those with clear cell histology, as now reported in the article (results section)

In the discussion, I would also recommend a paragraph or two about the non-ccRCC cohort.

A paragraph on non-ccRCC cohort has been added to the discussion

Some English editing is required. For example, Authors and Investigators do not need to be capitalized. Kg in line 76 does not need to be capitalized. There are also several awkward double spaces between two words throughout the text.

The revised version has been reviewed and checked for spelling mistakes and typing errors. Capitalized kg has been corrected. Double spaces  were accidentally added by the word Office program during adaptation to the CANCER template. They have been deleted.

Line 45 should say "Despite one third of patients not completing..."

The sentence has been corrected.

Make sure to define TKI before using it (line 481).

TKI abbreviation is now explained at its first appearance in the text (line XXX of revised version)

Reviewer 3 Report

Authors Basso et al have presented their work on combined immunotherapy in the setting of metastatic RCC patients over a period of 6-7 months. Their findings are meaningful in context to therapeutic intolerance and overall benefit to the patients in terms of OS. 

I have a few minor suggestions:

The introduction seems incomplete without addressing the burden of disease - a brief description of kidney cancer and epidemiology worldwide and in context to Italy would be advisable. There has to be explained the 3 main subtypes of kidney cancer and also why immunotherapy is beneficial in metastatic setting versus primary setting has not been addressed. 

Overall results do give a clear picture But I would suggest addressing some sub-points as well example- although brain metastasis patients are fewer - can we draw some meaningful conclusions about these patients? 

Similarly, it would have been interesting to see the response in clear cell RCC (CCRCC) versus non-Clear cell RCC (as primarily CCRCC are immune rich versus some like Chromophobe which is immune poor)- this is really meaningful in the clinical setting.

Also in recent years, there has been much interest in RCC cases with sarcomatoid differentiation due to inter tumor heterogeneity concept so I would recommend addressing this histopathological subtype in terms of overall survival and other parameters.  

The addition of these and similarly correlating these in discussion will tie up the manuscript nicely. In discussion, authors can in 2-3 lines discuss any differences/similarities they saw in their cohort and what has been reported in the literature regarding metastatic melanoma and other non kidney cancers. 

 starts very abruptly. 

Author Response

Authors Basso et al have presented their work on combined immunotherapy in the setting of metastatic RCC patients over a period of 6-7 months. Their findings are meaningful in context to therapeutic intolerance and overall benefit to the patients in terms of OS. 

I have a few minor suggestions:

The introduction seems incomplete without addressing the burden of disease - a brief description of kidney cancer and epidemiology worldwide and in context to Italy would be advisable. There has to be explained the 3 main subtypes of kidney cancer and also why immunotherapy is beneficial in metastatic setting versus primary setting has not been addressed. 

Epidemiology of renal cell carcinoma is now briefly presented in the Introduction, worldwide and in the context of Italy. The three major subtipes of kidney cancer are now mentioned. Recent results of adjuvant pembrolizumab in the localized setting of disease are now mentioned  in the last sentence of discussion.

Overall results do give a clear picture But I would suggest addressing some sub-points as well example- although brain metastasis patients are fewer - can we draw some meaningful conclusions about these patients? 

Your point is well taken. In fact, a detailed analysis of outcome of patients  according to site of disease (brain, liver, bone) is ongoing, but we think that the results deserve to be presented and discussed more properly in a separate publication.   

Similarly, it would have been interesting to see the response in clear cell RCC (CCRCC) versus non-Clear cell RCC (as primarily CCRCC are immune rich versus some like Chromophobe which is immune poor)- this is really meaningful in the clinical setting.

This point was raised also by reviewer 2. Patients with non-clear histology achieved lower OS, PFS and ORR compared to those with clear cell histology, as now reported in the article (results and discussion section).

Also in recent years, there has been much interest in RCC cases with sarcomatoid differentiation due to inter tumor heterogeneity concept so I would recommend addressing this histopathological subtype in terms of overall survival and other parameters.  

Fifty-six patients with sarcomatoid differentiation achieved  comparable PFS and ORR compared to the other patients, but overall survival was slightly lower, as now reported in the article and in the discussion.

The addition of these and similarly correlating these in discussion will tie up the manuscript nicely. In discussion, authors can in 2-3 lines discuss any differences/similarities they saw in their cohort and what has been reported in the literature regarding metastatic melanoma and other non kidney cancers.

The role of IPI-NIVO combination in metastatic melanoma is now  briefly mentioned and compared in terms of toxicity and toxic deaths in the discussion. 
